# Evaluation of the Chemical Composition of Selected Varieties of *L. caerulea* var. *kamtschatica* and *L. caerulea* var. *emphyllocalyx*

**DOI:** 10.3390/molecules28062525

**Published:** 2023-03-09

**Authors:** Józef Gorzelany, Oskar Basara, Ireneusz Kapusta, Korfanty Paweł, Justyna Belcar

**Affiliations:** 1Department of Food and Agriculture Production Engineering, University of Rzeszow, 4 Zelwerowicza Street, 35-601 Rzeszów, Polandjustyna.belcar@op.pl (J.B.); 2Doctoral School of the University of Rzeszów, st Rejtana 16C, 35-959 Rzeszów, Poland; 3Department of Food Technology and Human Nutrition, University of Rzeszow, 4 Zelwerowicza Street, 35-601 Rzeszów, Poland; ikapusta@ur.edu.pl; 4Fruit Plant Nursery ‘Korfanty’, 36-207 Grabownica Starzeńska, Poland; korfanty.farm@wp.pl

**Keywords:** *Lonicera*, chemical composition, quality fruits, organic acids, sugars, polyphenolic profile, antioxidant activity

## Abstract

*Lonicera caerulea* fruits are a rich source of vitamins, organic acids, and phenolic compounds, which are characterised by their health-promoting properties. The content of bioactive compounds in this fruit may vary depending on the cultivar and the harvest date. The fruits of the *L. caerulea* var. *kamtschatica* cultivars ‘Duet’ and ‘Aurora’ and the *L. caerulea* var. *emphyllocalyx* cultivars ‘Lori’, ‘Colin’ and ‘Willa’ were used in this study. *L. emphyllocalyx* fruit, especially the cultivar ‘Willa’, was characterised as having a higher acidity by an average of 29.96% compared to *L*. *kamtschatica*. The average ascorbic acid content of the *L*. *kamtschatica* fruit was 53.5 mg·100 g^−1^ f.w., while *L. emphyllocalyx* fruit had an average content that was 14.14% lower. The antioxidant activity (determined by DPPH, FRAP, and ABTS) varied according to the cultivar and the species of fruit analysed. The total polyphenol content differed significantly depending on the cultivar analysed; fruits of the *L. emphyllocalyx* cultivar ‘Willa’ were characterised by the lowest content of total polyphenols—416.94 mg GAE·100 g^−1^ f.w.—while the highest content of total polyphenols—747.85 GAE·100 g^−1^ f.w.—was found in the fruits of the *L. emphyllocalyx* cultivar ‘Lori’. *Lonicera caerulea* fruits contained 26 different phenolic compounds in their compositions, of which the highest content was characterised by cyanidin 3-*O*-glucoside (average: 347.37 mg·100 g^−1^). On the basis of this study, it appears that both *L*. *kamtschatica* fruits and *L. emphyllocalyx* fruits, especially of the cultivars ‘Lori’ and ‘Willa’, can be used in food processing.

## 1. Introduction

The blue honeysuckle (*Lonicera caerulea* L.) belongs to the genus *Lonicera* (*Caprifoliaceae*), which contains more than 200 species, native to the cold lands of the Far East and central Asia. Most of them are ornamental plants; only about 17 are edible fruit-producing species [1]. *Lonicera caerulea* has numerous varieties, several of which are widely cultivated, originating from Russia (*L*. *caerulea var*. *edulis*, *L*. *caerulea var*. *kamtschatica*, *L*. *caerulea var*. *altaica*, *L*. *caerulea var*. *boczkarnikovae*) and the island of Hokkaido in Japan (*L*. *caerulea* var. *emphyllocalyx*). They are long-living (25–30 years) shrubs that can reach 0.8–3.0 m height. They need an outside pollinator to bear fruits one year after planting. After three years, approx. 500 g of fruit can be obtained from one plant. Berries are dark blue or dark purple in colour, with a size of 1.5–3.0 cm, and a mostly cylindrical shape with a wax coating. They are plants that are very resistant to frost, able to withstand temperatures up to −46 °C [2]. *Lonicera caerulea* var. *kamtschatica* is a variety of honeysuckle, commonly known as Kamchatka berry, which is one of the most popular fruits in Poland, the Czech Republic, Canada, and Russia. Several varieties native to the species have been selected, which differ, among other things, in terms of flowering time, prolificacy, and content of bioactive substances [3]. A lesser-known variety is *Lonicera caerulea var*. *emphyllocalyx* (Maxim.) Nakai, commonly referred to as Japanese haskap. This is native to the island of Hokkaido in northern Japan and is also cultivated in China, Korea, or Russia [4]. The word ‘haskap’, from the Japanese hasukappu, hascup, haskappu, or hasakapu, in the language of the natives literally means a lot of small objects on the tops of the branches. In the regions where the plant originates, the fruit is recognised as a medicine and immune booster, used to treat stomach ailments and protect against many diseases. The native Ainu people of the island of Hokkaido called the fruit the ‘elixir of life’, making it an important part of their diet. They were eaten fresh, preserved with sugar and salt and were also used to prepare spirits and stockpile valuable raw material for winter [5,6]. 

The *L*. *caerulea* fruit is rich in sugars, organic acids, and polyphenols; this has a significant impact not only on sensory perception, but also on the health-promoting properties of the fruit. The content of polyphenols, the main group showing biological activity, can vary depending on the cultivar and the harvest date. The cultivation conditions (soil type and fertilisation method) do not significantly affect the content of active compounds; fruit from different sites will be characterised by similar contents of bioactive compounds [6]. *Lonicera caerulea* berries are characterised by a high content of phenolic compounds, ranging from 428.14 to 622.52 mg GAE·100 g^−1^ f.w. [7,8]. *L*. *caerulea* is an important source of phenolic compounds such as anthocyanins, flavonoids, proanthocyanidins and phenolic acids [6], the predominant group of phenolic compunds being anthocyanins, mainly cyanidin 3-*O*-glucoside. This is a widely used compound in the plant world, accounting for about 79–92% of the the anthocyanin content of *L*. *caerulea* fruit [9]. Fruits are a valuable source of vitamin C (ascorbic acid), at around 30.5–186.6 mg·100 g^−1^ [10]. The fruit is a source of minerals, including magnesium (79–163 mg·kg), potassium (3000–5000 mg·kg), phosphorus (486–2252 mg·kg), and calcium (1077 mg·kg) [11]. The berries are characterised by anti-inflammatory, antimicrobial, and antioxidant properties [6]. The fruits of *L*. *caerulea* has many uses in food processing. Pulp made from *L*. *caerulea* berries has a high juice yield and tiny seeds, making it excellent source of fruit juice [12]. Due to the very dense colour of the fruit juice, it is suitable for the production of food products that require the addition of juice. It also has high antioxidant properties [13,14]. Canned food and spreads are a relatively good way to improve the utility value of *L*. *caerulea*; however, due to the high content of sugars, the amount of added sugar should be controlled during the production stage [15]. The processing of fruits might change the levels of bioactive compounds. Fresh-processed foods have more health-promoting properties than the thermally treated foods. The concentration of some compounds might increase after water loss caused by fruit processing methods, e.g., heating. After freeze drying or pressing the contents of some compounds might change [15].

The purpose of this study was to compare the chemical compositions of the fruits of *L*. *caerulea* var. *kamtschatica* ( ‘Duet’ and ‘Aurora’ cultivars) and *L*. *caerulea* var. *emphyllocalyx* ( ‘Lori’, ‘Colin’, and ‘Willa’ cultivars) and the potential use of *L*. *emphyllocalyx* fruit in food processing.

## 2. Results and Discussion

### 2.1. pH and Acidity of Fruits

The organic acid content of the fruit decreases with successive stages of ripening. The organic acids contained in the fruit are perishable and, under the influence of various factors, e.g., temperature, can change in terms of concentration in the plant material [16,17]. *L*. *caerulea* fruit are rich in organic acids (e.g., malic acid, citric acid, quinic acid and fumaric acid; [17]). The contents of individual organic acids significantly influence the taste qualities of ripe fruit and the acceptability of the consumer. The average pH values of the *L*. *kamtschatica* and the *L*. *emphyllocalyx* fruits were similar: 3.07–3.32 and 3.13–3.52, respectively (Figure 1). 

These results are comparable to those obtained by other authors. In a study by Gerbrand et al. [18], the fruit pH of *Lonicera caerulea* ranged from 2.42 to 3.10; in an experiment of Auzanneau et al. [19], the fruit pH of the same species ranged from 2.70 to 3.30, depending on the growing year and harvest date. Compared to Miyashita et al. [20] and Oszmiański et al. [21], we found a lower average pH value in *L*. *kamtschatica* and *L*. *emphyllocalyx* berries: 2.60 and 2.65, respectively. The pH value is diversified compared to other species grown in Poland, e.g., saskatoon berry (4.12–5.03) [22], red currant (3.20 and 3.27) [23], sea buckthorn (3.02–3.19) [24], highbush blueberry fruit (2.76–3.33) [25], raspberry (3.72) and mulberry (5.17) [26]. In general, both *L*. *kamtschatica* and *L*. *emphyllocalyx* are rich in organic acids that give a taste resembling blueberries, with a distinct hint of sourness.

The difference in acidity between varieties may have been caused by different climatic conditions. The acidity of the *L*. *kamtschatica* fruit was at the level of 1.22–1.31 g·100 g^−1^, while the *L*. *emphyllocalyx* fruit, especially the ‘Willa’ cultivar, had a higher acidity by 29.96% on average compared to the *L*. *kamtschatica* fruit (Figure 1). The results in this study are similar to a study by MacKenzie et al. [27], where the total acidity in *L*. *caerulea* fruit ranged from 1.6 to 3.0 g·100 g^−1^, depending on the year of cultivation. The obtained results are comparable with berries grown in Poland, e.g., cranberry (1.56–1.60 g·100 g^−1^) [28], saskatoon berry (0.3–1.5 g·100 g^−1^) [22], red currant (0.7–1.6 g·100 g^−1^) [29], mulberry fruit (0.26 g·100 g^−1^) and raspberry (0.63 g·100 g^−1^) [26].

### 2.2. Contents of Ascorbic Acid and Antioxidant Activity in L. kamtschatica and L. emphyllocalyx Fruit 

Ascorbic acid is a powerful antioxidant that is required for the activation of many enzymes. This compound is essential for the functioning of the human body, influencing the immune and circulatory systems, accelerating wound healing, slowing skin ageing, and regulating collagen production [30]. The average ascorbic acid content of the *L*. *kamtschatica* fruit was 53.5 mg·100 g^−1^ f.w., while the *L*. *emphyllocalyx* fruit had an average ascorbic acid content of 45.93 mg·100 g^−1^ f.w. (Table 1). The content ascorbic acid of *L*. *caerulea* fruit differs from that reported by Celli et al. [10]; we obtained results ranging from 30.5 to 186.6 mg·100 g^−1^ f.w. According to Jurnikova et al. [31], the content of ascorbic acid in *Lonicera caerulea*, higher than 70 mg·100 g^−1^ f.w., is associated with a lower accumulation of phenolics. The ascorbic acid content of *L*. *kamtschatica* and *L*. *emphyllocalyx* fruit was at a similar level to popular berries grown in Poland, e.g., strawberries (average: 50.1 mg·100 g^−1^), raspberries (average: 30.6 mg·100 g^−1^ f.w.), blueberries (average: 60.1 mg·100 g^−1^ f.w.) [32,33], plum blackthorn (21.94 mg·100 g^−1^ f.w.), blackberry (33.85 mg·100 g^−1^ f.w.) [33]; red currant (31.2 mg·100 g^−1^–44.1 mg·100 g^−1^ f.w.) [23], sea buckthorn (4.0–9.1 mg·100 g^−1^ f.w.) [24]. According to Senica et al. [14], spreads and smoothies made out of *L*. *caerulea* berries increase the concentration of ascorbic acid by 100%. Heating plant material before crushing may influence the stability of ascorbic acid, preserving its contents. This can be used in the food industry in the production of *L*. *caerulea* products [14].

*Lonicera caerulea* fruit extract containing anthocyanins is a highly effective antioxidant agent, effectively reducing reactive oxygen species (ROS), which are produced by immune cells as a result of inflammation. Persistent inflammation can lead to cell damage and chronic disease. The fruit extract further reduces lipid peroxidation, which affects cellular damage under oxidative stress conditions, which can contribute to the reduction in diseases related to oxidative stress [34]. The antioxidant activity of *L*. *kamtschatica* and *L*. *emphyllocalyx* fruit, determined by the DPPH method, was at the level of 68.68–89.62% inhibition, of which the *L*. *kamtschatica* cultivar ‘Duet’ had the highest free radical scavenging activity (on average 13.4–30.4% more than the other cultivars analysed). These results are comparable to those obtained by Kula et al. [2] and Khattab et al. [35], in which the antiradical values of *Lonicera caerluea* were, respectively, 85% and 78.70%. The antioxidant activity of *L*. *caerulea* fruits is significantly higher compared to other species, e.g., sea buckthorn (74%), bilberry (37–91%), and garden rhubarb (48–98%) [36]. The iron reduction capacity (FRAP method) of *L*. *kamtschatica* range from 30.52 to 37.67 μM Fe^2+^·g^−1^ f.w. and *L*. *emphyllocalyx* range from 30.52 to 37.67 μM Fe^2+^·g^−1^ f.w. Results comparable to this study were obtained by Rupasinghe et al. [7], with an antioxidant value (FRAP) ranging from 27.96 to 46.90 μM Fe^2+^·g^−1^. According to studies, *L*. *caerulea* had more antioxidative properties (FRAP) compared to other species grown in Poland, e.g., strawberries (8.00 µM TE·g^−1^ f.w), blackberries (15.03 µM TE·g^−1^ f.w.), highbush blueberry (16.24 µM TE·g^−1^ f.w.), elderberry (29.56 mM Fe^2+^·g^−1^ f.w.), blackthorn (14.74 mM Fe^2+^·g^−1^ f.w.) and wild strawberry (10.99 Fe^2+^·g^−1^ f.w.) [7,33]. *Lonicera caerulea* fruits are characterised by a higher antioxidant capacity. The antioxidant activity of the fruits of *L*. *kamtschatica* and *L*. *emphyllocalyx*, determined by the ABTS method, ranged from 1.91 to 2.21 mM TE·100 g^−1^ f.w. As reported by Rop et al. [37], the ABTS antiradical activity of ABTS of *Lonicera caerulea* was 0.30 µM·g^−1^. The values obtained in this study are different to other species cultivated in Poland, e.g., highbush blueberry (16.87 mM TE·g^−1^ f.w.), elderberry (15.88 mM TE·g^−1^ f.w.), blackberry (9.55 mM TE·g^−1^ f.w.), strawberries (5.61 TE·g^−1^ f.w.) [7,33], red currant (11.83–12.59 µM TE·g^−1^ d.m.) [13], mulberry (0.14 mM TE·100 g^−1^ f.w.) and raspberry (0.08 mM TE·100 g^−1^ f.w.) [26]. The high level of antioxidant capacity in *L*. *kamtschatica* and *L*. *emphyllocalyx* makes them very valuable in terms of bioactivity. Research on the antioxidant activity of extracts with *L*. *caerulea* revealed that the fruits of this plant are characterised by stronger antioxidant properties from other berries commonly regarded as effective antioxidants. Considering the relationship between modern-day diseases and long-term oxygen stress, strong antioxidant properties may indicate the potential importance of this fruit, not only in prophylaxis, but also in the treatment of many diseases [2], thus making *L*. *caerulea* products more valuable for food industry.

### 2.3. Polyphenolic Compound Content in L. kamtschatica and L. emphyllocalyx Fruit

The phenolic compounds, mainly anthocyanins, contained in *L*. *caerulea* fruit extract exhibit anti-inflammatory effects. They reduce cellular damage under conditions of oxidative stress in in vitro cultures of rat microsomes and reduce ROS production in cultures of pro-inflammatory gingival fibroblasts [6,38]. The contents of total polyphenols differed significantly depending on the cultivar analysed; fruits of the *L*. *emphyllocalyx* cultivar ‘Willa’ were characterised by the lowest total polyphenol content—416.94 mg GAE·100 g^−1^ f.w.—while the highest total polyphenol content of total polyphenols—747.85 GAE·100 g^−1^ f.w.—was also found in fruits of *L*. *emphyllocalyx* but of the cultivar ‘Lori’ (Table 2). The results obtained in this study are comparable to those reported by Rop et al. [37]; the content of total polyphenols in the *L*. *kamtschatica* fruit ranged from 575 to 903 mg GAE·100 g^−1^ f.w, while in the study by Oszmiański et al. [21], the polyphenol content of the *L*. *kamtschatica* fruit was 12.29 g·100 g^−1^ f.w. The analysed fruits of *L*. *emphyllocalyx* and *L*. *kamtschatica* were characterised by significantly higher total polyphenol contents compared to fruits grown in Poland: raspberries—445.5 mg GAE·100 g^−1^ f.w.; strawberries—238.0 mg GAE·100 g^−1^ f.w.; sea buckthorn—302.72 mg GAE·100 g^−1^ f.w. [6,39,40]; blackberry—247.25 mg GAE·100 g^−1^ f.w.; blackthorn—402.67 mg GAE·100 g^−1^ f.w.; highbush blueberry—424.72 mg GAE·100 g^−1^ f.w.; and elderberry—535.98 mg GAE·100 g^−1^ f.w. [33]. 

The analysis of phenolic compounds using the UPLC-PDA-MS/MS method allowed for the determination of the differences between the contents of individual groups of polyphenolic compounds contained in the fruit of the analysed cultivars of *L*. *kamtschatica* and *L*. *emphyllocalyx* (Table 2). Berries of the *L*. *kamtschatica* cultivars ‘Duet’ and ‘Aurora’, and *L*. *emphyllocalyx* cultivars ‘Lori’, ‘Willa’ and ‘Colin’ were characterised by different contents of individual polyphenolic compounds. The total content of phenolic compounds depends, among other things, on the cultivar, the degree of fruit maturity, and also on the harvest date. The identification of compounds was carried out based on retention time (Rt), MS and MS/MS with available publications [41,42,43,44,45]. The extract prepared from the fruit contained 26 different compounds in its composition, including 14 anthocyanins, 8 flavonoids, 2 phenolic acids and 1 flavan-3-ol (Table 2).

The highest anthocyanin content in the analysed fruits was found in the berries *of L*. *kamtschatica* ‘Duet’, at 456.3 mg·100 g^−1^ (Table 2). Among the phenolic compounds found in the fruits of *L*. *kamtschatica* (509.29–597.29 mg GAE·100 g^−1^) and *L*. *emphyllocalyx* (416.94–747.85 mg GAE·100 g^−1^), anthocyanins represented, on average, 94% of all polyphenols, and the main representative was cyanidin 3-*O*-glucoside—C3G (the compound represented, on average, 82.2% of the total anthocyanin content detected in the fruits). Among the varieties analysed, the *L*. *kamtschatica* fruits of the ‘Duet’ variety contained the highest concentration of C3G in their composition—382.18 mg·100 g^−1^ (Table 2). The results obtained in this study are comparable to those obtained by Rupasinghe et al. [9]; the C3G content of *Lonicera caerulea* fruit ranged from 68 to 649 mg·100 g^−1^. C3G content was comparable to that reported by Khattab et al. [35], and the C3G content in the fruit of the cultivars ‘Tundra’, ‘Berry Blue’ and ‘Indigo gem’ reached 79–88% of the total anthocyanin content. The C3G content of the *Lonicera caerulea* fruit was significantly higher compared to the strawberry fruit (3.7 mg·100 g^−1^), blueberry (3.0 mg·100 g^−1^), and the cranberry (0.7 mg·100 g^−1^) [9]. C3G is the most prevalent anthocyanin in edible fruits and has been shown to have anti-inflammatory, antioxidant, chemotherapeutic, and epigenetic effects [9]. Fruits with a larger diameter and harvested at the optimal harvest time will have a higher anthocyanin content due to the larger skin area [15]. In the study of Senica et al. [15], spreads and smoothies made out of *L*. *caerulea* had higher concentrations of C3G (6.48, 5.00 mg·100 g^−1^) in comparison to fresh fruit. The fruits of this species play important roles in a wide range of physiological processes, e.g., protective effect against UV radiation for skin, protection against pathogenic strains, etc..

The total flavonoid contents ranged from 11.11 mg·100 g^−1^ in the *L*. *emphyllocalyx* cv. cultivar ‘Colin’ to 24.4 mg·100 g^−1^ in the case of the *L*. *kamtschatica* cv. cultivar ‘Duet’ (Table 2). In the *L*. *kamtschatica* and *L*. *emphyllocalyx* fruits analysed, the average content of flavonoids accounted for 3% of all phenolic compounds and the main representative was quercetin 3-*O*-rutinoside, which represented an average of 41.6% of all flavonoids (Table 2). Quercetin 3-*O*-rutinoside content in our study was significantly higher compared to the results of Oszmiański et al. [21]; the *L*. *kamtschatica* fruits of the cultivar ‘Wojtek’ were characterised by a quercetin 3-*O*-rutinoside content of 0.21 mg·100 g^−1^ f.w. Quercetin 3-*O*-rutinoside shows a protective effect on the liver or blood vessels and has anti-inflammatory and antidiabetic properties [46].

Two phenolic acids, chlorogenic acid and neochlorogenic acid, were determined in the composition of the *L*. *kamtschatica* and *L*. *emphyllocalyx* fruits. The fruits of *L*. *kamtschatica* cultivars were characterised by a higher content of phenolic acids than those of *L*. *emphyllocalyx* cultivars. Of the *L*. *kamtschatica* and *L*. *emphyllocalyx* cultivars analysed, the *L*. *kamtschatica* cultivar ‘Duet’ was characterised by the highest contents of phenolic acids, on average 14.62 mg·100 g^−1^, which was 51% higher than in the fruit of the *L*. *emphyllocalyx* cultivar ‘Colin’ (Table 2). Chlorogenic acid, with an average of 9.77 mg·100 g^−1^, was present in the analysed fruits at a significantly higher concentration than neochlorogenic acid (Table 2). The chlorogenic acid contained in the fruits analysed constituted 91.8% of all phenolic acids, which is comparable to the results of Kithama et al. [47], who determined that, in the cultivars ‘Aurora’, ‘Evie’, ‘Larissa’ and ‘Rebecca’, the content of chlorogenic acid made up 95% of all phenolic acids contained in *L*. *kamtschatica* fruits. 

A chemical compound from the flavan-3-ol group—B-type procyanidins—was present among the fruits tested of *L*. *kamtschatica* and *L*. *emphyllocalyx* cultivars. The highest concentration of B-type procyanidins was determined in the berries of the *L*. *emphyllocalyx* cultivar ‘Willa’—an average of 2.35 mg·100 g^−1^, which was on average 62.1% higher than in the fruit of the cultivar ‘Colin’ (Table 2). In a study by Raudonė [48], *L*. *kamtschatica* fruits of the cultivars ‘Wojtek’, ‘Indigo Gem’, ‘Iga’, ‘Leningradskij Velikan’, ‘Nimfa’, ‘Amphora’, ‘Tola’ and ‘Tundra’ contained from 9.76 mg·100 g^−1^ f.w. to 27.1 mg·100 g^−1^ f.w. of procyanidin type B procyanidin in their compositions. *L*. *kamtschatica* fruits of the cultivar ‘Wojtek’ in the study by Oszmiański et al. [21] were characterised by a procyanidin dimer content of 7.05 mg·100 g^−1^ f.w.

### 2.4. Sugar Content of L. kamtschatica and L. emphyllocalyx Fruit

Of the *L*. *kamtschatica* berry cultivars analysed, the ‘Duet’ cultivar had the highest total sugar content, with an average of 4968 mg·100 g^−1^, and glucose was the dominant sugar in the fruit of all the cultivars (Table 3). A significant difference in the contents of total sugars was observed between the fruits of the *L*. *kamtschatica* and the *L*. *emphyllocalyx* cultivars. The *L*. *kamtschatica* fruits of the cultivar ‘Aurora’ were characterised by the highest glucose content, on average 2963 mg·100 g^−1^, while the significantly lower glucose content between the analysed cultivars was characteristic of *L*. *emphyllocalyx* fruits of the cultivar ‘Willa’ (Table 3). However, this cultivar was characterised by a high fructose content (comparable to that of the *L*. *kamtschatica* cultivar ‘Duet’) and significantly higher compared to the other cultivars (Table 3). The sucrose content of the fruits analysed ranged from 1325 mg·100 g^−1^ (*L*. *emphyllocalyx* cultivar ‘Willa’) to 2750 mg·100 g^−1^ (*L*. *kamtschatica* cultivar ‘Aurora’; Table 3). The sugar content results are comparable to those obtained by Senica et al. [49], who obtained a total sugar content ranging from 1557.37 to 2585.45 mg· 100 g^−1^ f.w. The results obtained by Gołba et al. [11] were significantly lower compared to those of this study, the total sugar content of the *Lonicera caerulea* fruit ranged from 1500 mg·100 g^−1^ to 2585 mg·100 g^−1^. The average fructose content of *Lonicera caerulea* reported by Sharma et al. [17] was comparable with this study and ranged from 1047.53 to 1363.67 mg·100 g^−1^ f.w., but the content of glucose was much smaller and ranged from 750.89 to 1129.35 mg·100 g^−1^ f.w. A range of 3.72–126.12 mg·100 g^−1^ f.w. of sucrose has also been reported by Cheng et al. [50]; these values are comparable to those obtained in the experiment. As reported by Wojdyło et al. [51], Sorbitol was previously found in Polish *L*. *kamtschatica* cultivars, with a concentration ranging from 0.1 to 0.4 mg·100 g^−1^ f.w. The sugar content of *L*. *kamtschatica* and *L*. *emphyllocalyx* fruit depends on environmental conditions, light intensity, fruit maturity and species, among other factors [16].

## 3. Materials and Methods

### 3.1. Material

Fruits of the *L*. *caerulea* var. *kamtschatica* cultivars ‘Duet’ and ‘Aurora’ were obtained from a nursery crop located in Tyczyn (49°57′52″ N 22°2′47″ E, Subcarpathian Voivodship, Poland) in the year 2022. Fruits of the *L. caerulea* var. *emphyllocalyx* cultivars ‘Lori’, ‘Willa’ and ‘Colin’ were obtained from ‘Korfanty’ (49°41′41″ N 22°5′3″ E, Grabownica Starzeńska, Subcarpathian Voivodeship, Poland) in 2022. Both species were grown in containers filled with a peat substrate containing sand and perlite in a ratio of 20:1:1, with the addition of fertilizer Osmocote Exact 3–4 m (ICL, Sydney, Autralia) at concentration of 2.0 kg for 1 m^3^ of substrate.

The average monthly temperatures in the period from March to June in Tyczyn were, respectively, 3.3, 7.0, 14.8, and 19.8 °C, and in Grabownica Starzeńska were, respectively, 2.2, 6.0, 13.7, and 18.5 °C. The average monthly rainfall values in the period from March to June in Tyczyn were, respectively, 20 mm, 60 mm, 50 mm, and 20 mm, and in Grabownica Starzeńska the average monthly rainfall in the period from March to June was 50 mm.

The fruits of the analysed cultivars were harvested by hand at the stage of their harvest maturity (first decade of June), 1000 g each. Immediately after harvest, the fruits were subjected to chemical analysis. 

### 3.2. Determination of pH and Acidity

The total acidity (as citric acid) and the pH of the *L*. *kamtschatica* and *L*. *emphyllocalyx* fruit were determined through the potentiometric titration of the sample for analysis with a standard 0.1 M NaOH solution at pH = 8.1 using a TitroLine 5000 (SI Analytcs, Weilheim, Germany) according to the method given in PN-EN 12147:2000 [52]. The results are expressed as g of citric acid per 100 g of fruit. The analyses were performed in triplicate. 

### 3.3. Determination of the Contents of Bioactive Compounds in Fruit and Determination of Their Antioxidant Activity

Vitamin C (ascorbic acid) was determined according to PN-A-04019:1998 [53]. Total polyphenol content (mg GAE·100 g^−1^ f.w.) was determined using the Folin–Ciocalteu method according to the methodology described by Bakowska-Barczak et al. [8]. The identification of the polyphenolic profile in *L*. *kamtschatica* and *L*. *emphyllocalyx* fruit was determined according to the methodology reported by Gorzelany et al. [24]. 

The ability of the fruit to reduce iron ions (FRAP method) was determined according to the methodology given by Rupasinghe et al. [26], and the results are given in μM Fe^2+^·g^−1^ f.w. The antioxidant activity of the fruit was determined using DPPH methods according to the methodology given by Jurčaga et al. [54], and the result is expressed as % inhibition of DPPH radicals, and through the ABTS method according to the methodology given by Gawroński et al. [1], and the results are expressed in mM TE·100 g^−1^ f.w. All analyses were performed in triplicate. 

### 3.4. Determination of Sugars in L. kamtschatica and L. emphyllocalyx Fruit 

The sugar content was measured using the HPLC method with refractive index detection. The chromatographic equipment SYKAM (Sykam GmbH, Eresing, Germany), consisting of sample injector S5250, pump system S1125, column oven S4120 and RI detector S3590, was used. Separation was carried out using a Cosmosil Sugar-D column 250 × 4.6 mm ( Nacalai, Kyoto, Japan). The separation was achieved with a mobile phase of 70% of acetonitrile in water in isocratic mode. The flow rate was 0.5 mL/min at column temperature set at 30 °C. The volume of injected sample was 20 µL and 15 min was needed to complete the analysis. Samples before injection were centrifuged at 5000 rpm for 10 min using Centrifuge 5430 (Eppendorf, Hamburg, Germany) and diluted with mobile phase 1:4 (*v*/*v*). All determinations were performed in triplicate. 

### 3.5. Statistical Analysis 

Using Statistica 13.3. software (TIBCO Software Inc., Tulsa, OK, USA), a statistical analysis of the results obtained was performed that included the analysis of variance (ANOVA) and NIR significance test at a significance level of α = 0.05. 

## 4. Conclusions 

Based on this study, differences in fruit composition were found in both individual species and the *L*. *caerulea* cultivars. The fruits of the *L*. *kamtschatica* cultivar ‘Aurora’ contained the highest amount of ascorbic acid in their composition, approximately 22% more compared to ‘Lori’, which had the highest content of this compound among *L*. *emphyllocalyx* cultivars. The highest total polyphenol content was found in the *L*. *emphyllocalyx* cultivar ‘Lori’, while the predominant polyphenolic compound was cyanidin 3-*O*-glucoside. On the basis of the results obtained, it can be concluded that the *L*. *emphyllocalyx* cultivars ‘Lori’ and ‘Willa’ can find applications in various food industries. Despite the diversity of chemical composition, the fruits have high antioxidant properties compared to the well-known varieties ‘Duet’ and ‘Aurora’. The variety ‘Lori’ has the highest content of phenolic compounds among the tested varieties. ‘Willa’ has the highest concentration of C3G comparable to the widely known varieties of *L*. *kamtschatica* and the highest sugar content among *L*. *emphyllocalyx*. The use of *Lonicera* as an ingredient of functional foods, natural colourant or source of natural antioxidant seems to be promising. Compared to other types of berries grown in Poland, e.g., raspberries and red currant, *L*. *kamtschatica* and *L*. *emphyllocalyx* can be a good source of bioactive substances and sugar, making them good in food industry and processing, including juices, wines, spreads and dried fruits.

## Figures and Tables

**Figure 1 molecules-28-02525-f001:**
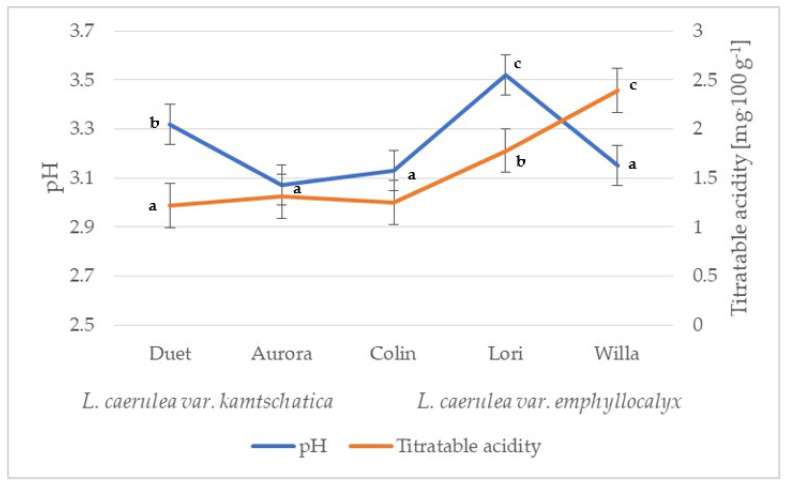
pH and total acidity of *L. caerulea* var. *kamtschatica* and *L. caerulea* var. *emphyllocalyx* fruits (data are expressed as mean values (*n* = 3) ± SD; SD—standard deviation). Mean values with different letters are significantly different (*p* < 0.05).

**Table 1 molecules-28-02525-t001:** Ascorbic acid content and antioxidant activity of *L. kamtschatica* and *L. emphyllocalyx* fruit.

		*Lonicera kamtschatica*		*Lonicera emphyllocalyx*		
		‘Duet’	‘Aurora’	‘Colin’	‘Lori’	‘Willa’
Ascorbic acid content [mg·100 g^−1^ f.w.]		44.40 ^b^ ± 0.4	62.60 ^e^ ± 0.6	49.70 ^c^ ± 0.7	51.30 ^d^ ± 0.3	36.80 ^a^ ± 0.5
Antioxidant activity	DPPH [% inhibition]	89.62 ^d^ ± 0.01	68.68 ^a^ ± 0.08	72.11 ^b^ ± 0.01	79.02 ^b^ ± 0.02	68.97 ^a^ ± 0.04
	FRAP [μM Fe^2 +^·g^−1^ f.w.]	37.67 ^e^ ± 0.07	30.52 ^a^ ± 0.02	36.77 ^d^ ± 0.07	33.17 ^b^ ± 0.07	35.22 ^c^ ± 0.02
	ABTS [mM TE·100 g^−1^ f.w.]	2.26 ^b^ ± 0.06	1.97 ^a^ ± 0.07	1.91 ^a^ ± 0.01	2.12 ^b^ ± 0.02	2.21 ^c^ ± 0.04

Data are expressed as mean values (*n* = 3) ± SD; SD—standard deviation. Mean values within rows with different letters are significantly different (*p* < 0.05).

**Table 2 molecules-28-02525-t002:** Individual phenolic compounds identified by UPLC-PDA-MS/MS in *L*. *kamtschatica* i *L. emphyllocalyx*.

No.	Compund [mg·100 g^−1^]	Rt	λ_max_	[M-H]^+/−^ *m*/*z*		*Lonicera emphyllocalyx*			*Lonicera kamtschatica*	
		min	nm	MS	MS/MS	‘Lori’	‘Colin’	‘Willa’	‘Aurora’	‘Duet’
Anthocyanins										
1.	(+)Catechin-Cyanidin-3-*O*-glucoside	1.99	280, 522	737^+^	449, 287	1.30 ^a^ ± 0.10	1.21 ^a^ ± 0.04	1.35 ^a^ ± 0.31	1.56 ^a^ ± 0.23	1.41 ^a^ ± 0.21
2.	Pelargonidin 3-O-rutinoside	2.01	279, 520	579^+^	271	1.12 ^bc^ ± 0.10	0.87 ^ab^ ± 0.05	0.67 ^a^ ± 0.03	0.63 ^a^ ± 0.44	1.30 ^c^ ± 0.31
3.	(−)Epicatechin-Cyanidin-3-*O*-glucoside	2.08	279, 520	737^+^	449, 287	1.67 ^b^ ± 0.22	1.05 ^a^ ± 0.01	1.14 ^a^ ± 0.19	1.18 ^a^ ± 0.14	2.09 ^b^ ± 0.48
4.	Cyanidin 3,5-*O*-diglucoside	2.20	276, 512	611^+^	287	26.31 ^c^ ± 1.48	17.50 ^ab^ ± 0.56	20.40 ^bc^ ± 2.42	14.00 ^a^ ± 1.54	40.49 ^d^ ± 6.62
5.	Malvidin 3-*O*-glucoside	2.33	279, 522	493^+^	331	1.70 ^d^ ± 0.27	1.36 ^bc^ ± 0.10	1.05 ^b^ ± 0.20	0.72 ^a^ ± 0.10	1.47 ^cd^ ± 0.14
6.	Cyanidin 3-*O*-glucoside	2.61	279, 515	449^+^	287	355.73 ^b^ ± 8.40	259.41 ^a^ ± 2.03	364.89 ^b^ ± 33.45	374.62 ^b^ ± 18.00	382.18 ^b^ ± 28.34
7.	Cyanidin 3-*O*-rutinoside	2.93	279, 517	595^+^	287	33.73 ^d^ ± 0.20	32.06 ^cd^ ± 1.82	30.67 ^c^ ± 3.07	16.06 ^b^ ± 1.00	12.48 ^a^ ± 0.45
8.	Cyanidin 3-*O*-galactoside	3.05	278, 521	449^+^	287	0.12 ^a^ ± 0.03	0.82 ^d^ ± 0.09	0.63 ^c^ ± 0.06	0.37 ^b^ ± 0.01	0.38 ^b^ ± 0.01
9.	Pelagonidin 3-*O*-glucoside	3.16	277, 502	433^+^	271	1.40 ^a^ ± 0.04	2.05 ^b^ ± 0.13	2.23 ^b^ ± 0.37	3.24 ^c^ ± 0.27	1.28 ^a^ ± 0.14
10.	Pelargonidin 3-*O*-galactoside	3.28	277, 505	433^+^	271	0.32 ^b^ ± 0.04	0.83 ^c^ ± 0.04	0.38 ^b^ ± 0.03	0.76 ^c^ ± 0.01	0.19 ^a^ ± 0.06
11.	Peonidin 3-*O*-glucoside	3.46	279, 517	463^+^	301	21.42 ^c^ ± 0.55	14.27 ^ab^ ± 0.06	16.15 ^b^ ± 2.00	16.29 ^b^ ± 1.56	12.57 ^a^ ± 0.40
12.	Cyanidin 3-(6”-acetylo)-glucoside	3.59	279, 519	591^+^	449, 287	2.52 ^c^ ± 0.02	2.75 ^d^ ± 0.17	1.69 ^b^ ± 0.10	0.79 ^a^ ± 0.04	0.63 ^a^ ± 0.06
13.	Cyanidin 3-*O*-rutinoside-5-*O*-dlucoside	3.79	279, 522	757^+^	595, 287	0.13 ^b^ ± 0.03	0.15 ^b^ ± 0.03	0.43 ^c^ ± 0.05	0.39 ^c^ ± 0.02	0.07 ^a^ ± 0.01
14.	Delphinidin 3-*O*-glucoside-pentoside	3.77	279, 522	597^+^	303	0.26 ^c^ ± 0.07	0.16 ^b^ ± 0.00	0.26 ^c^ ± 0.01	0.37 ^d^ ± 0.01	0.09 ^a^ ± 0.02
Phenolic acids										
15.	Neochlorogenic acid	2.25	288sh, 324	353^−^	191	0.68 ^b^ ± 0.02	0.42 ^a^ ± 0.01	0.76^b^ ± 0.10	1.18^c^ ± 0.01	1.28^d^ ± 0.02
16.	Chlorogenic acid	2.87	288sh, 324	353^−^	191	6.92 ^a^ ± 0.26	8.82 ^b^ ± 0.48	9.94^b^ ± 1.41	13.44^c^ ± 0.39	9.71^b^ ± 0.81
Flavon-3-ols										
17.	Procyanidin dimer B-type	3.11	279	577^−^	289	2.10 ^b^ ± 0.25	1.46 ^a^ ± 0.05	2.35 ^b^ ± 0.22	1.97 ^b^ ± 0.10	2.32 ^b^ ± 0.36
Flavonoids										
18.	Quercetin 3-*O*-rutinoside-7-*O*-rhamnoside	4.03	255, 350	755^−^	609, 301	0.30 ^b^ ± 0.00	0.45 ^c^ ± 0.02	0.30 ^b^ ± 0.02	0.56 ^d^ ± 0.01	0.21 ^a^ ± 0.04
19.	Quercetin 3-*O*-arabinoside-glucoside	4.22	255, 355	595^−^	301	4.44 ^e^ ± 0.20	3.06 ^d^ ± 0.06	0.88 ^a^ ± 0.16	2.02 ^c^ ± 0.07	1.12 ^b^ ± 0.01
20.	Quercetin 3-*O*-rutinoside	4.54	255, 355	609^−^	301	3.04 ^a^ ± 0.09	2.65 ^a^ ± 0.15	4.75 ^b^ ± 0.66	14.08 ^c^ ± 0.29	5.32 ^b^ ± 0.11
21.	Quercetin 3-*O*-glucoside	4.72	255, 355	463^−^	301	1.88 ^c^ ± 0.02	1.32 ^b^ ± 0.02	0.98 ^a^ ± 0.15	2.46 ^d^ ± 0.25	1.92 ^c^ ± 0.27
22.	Quercetin 3-*O*-pentoside	4.88	255, 355	433^−^	301	0.15 ^a^ ± 0.01	0.20 ^ab^ ± 0.03	0.17 ^a^ ± 0.07	0.29 ^b^ ± 0.05	0.87 ^c^ ± 0.11
23.	Quercetin 3-*O*-rhamnoside	3.95	255, 355	447^−^	301	0.11 ^a^ ± 0.01	0.43 ^c^ ± 0.06	0.23 ^b^ ± 0.04	0.53 ^d^ ± 0.04	0.44 ^c^ ± 0.02
24.	Kaempferol 3-*O*-rutinoside	5.09	265, 347	593^−^	285	0.06 ^a^ ± 0.00	0.07 ^a^ ± 0.01	0.08 ^a^ ± 0.02	0.41 ^b^ ± 0.05	0.04 ^a^ ± 0.02
25.	3,4–di-*O*-caffeoyl-quinic acid	5.25	288sh, 324	515^−^	353	1.90 ^b^ ± 0.15	1.87 ^b^ ± 0.08	3.23 ^c^ ± 0.38	2.92 ^c^ ± 0.02	1.22 ^a^ ± 0.02
26.	Quercetin 3-O-(6”-acetylo)-glucoside	5.50	255, 333	505^−^	463, 301	0.92 ^b^ ± 0.01	1.06 ^c^ ± 0.05	0.66 ^a^ ± 0.06	1.11 ^c^ ± 0.04	0.89 ^b^ ± 0.13
Total polyphenols content [mg GAE·100 g f.w.]						747.85 ^e^ ± 0.05	522.06 ^c^ ± 0.06	416.94 ^a^ ± 0.04	597.29 ^d^ ± 0.04	506.29 ^b^ ± 0.06

Data are expressed as mean values (*n* = 3) ± SD; SD—standard deviation. Mean values within rows with different letters are significantly different (*p* < 0.05).

**Table 3 molecules-28-02525-t003:** Total sugar content of the fruits of *L. kamtschatica* and *L. emphyllocalyx*.

	*Lonicera kamtschatica*		*Lonicera emphyllocalyx*		
	‘Aurora’	‘Duet’	‘Lori’	‘Colin’	‘Willa’
Fructose content [mg·100 g^−1^]	1503.00 ^b^ ± 5	2109.00 ^c^ ± 2	1179.00 ^a^ ± 9	1163.00 ^a^ ± 4	2087.00 ^c^ ± 4
Glucose content [mg·100 g^−1^]	2963.00 ^c^ ± 3	2666.00 ^b^ ± 1	2943.00 ^c^ ± 5	2795.00 ^bc^ ± 5	2125.00 ^a^ ± 3
Sucrose content [mg·100 g^−1^]	275.00 ^c^ ± 4	192.00 ^b^ ± 6	192.00 ^b^ ± 1	171.00 ^b^ ± 5	132.00 ^a^ ± 5
Total sugar content [mg·100 g^−1^]	4741.00 ^bc^ ± 0.4	4968.50 ^c^ ± 0.4	4315.50 ^ab^ ± 0.8	4129.50 ^a^ ± 0.1	4345.50 ^ab^ ± 0.9

Data are expressed as mean values (*n* = 3) ± SD; SD—standard deviation. Mean values within rows with different letters are significantly different (*p* < 0.05).

## Data Availability

Not applicable.

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
