# Peer review of "Evaluation of the Chemical Composition of Selected Varieties of L. caerulea var. kamtschatica and L. caerulea var. emphyllocalyx"

_molecules, 2023, doi:10.3390/molecules28062525_

Round 1

Reviewer 1 Report

It is a complex work, with possible applications, but that can be improved as follows:

The introduction can be improved introducing more recently papers about L. caerulea composition and its applications in food industry.

In the introduction part and conclusions section mention that L. caerulea fruits can be used in food industries, but the possibilities to use these in food processing is not presented in "Results and discussion" part. Please correlate the composition of studied fruits with their potential use in food industry. Which could be the advantage of their using compared to other fruits?

In the title of section 2.2. "ascorbic acid" must be used instead of "bioactive compounds".

Please use in the entire manuscript the same abbreviations for the same notions: "f.w." or "f.m." (e.g. in table 1 appears "f.w." and in other parts of manuscript it is used "f.m.").

In the table 2, the column with Rt (retention time) appear the same values from the next column (wavelength). Please introduce the correct values.

Author Response

Authors are grateful for the contribution of the Reviewer

Point 1. The introduction can be improved introducing more recently papers about L. caerulea composition and its applications in food industry.

Response 1. The introduction was supplemented with literature on application in food industry – Lines: 75 – 86

Point 2. In the introduction part and conclusions section mention that L. caerulea fruits can be used in food industries, but the possibilities to use these in food processing is not presented in "Results and discussion" part. Please correlate the composition of studied fruits with their potential use in food industry. Which could be the advantage of their using compared to other fruits – Line 74 – 85, 424 -429

Response 2. The Results and discussions were supplemented with literature concerning use in food processing – Lines: 162 - 165, 202 – 210, 270 – 274,

Point 3. In the title of section 2.2. "ascorbic acid" must be used instead of "bioactive compounds".

Response 3.  The title remains the same, paragraph concerns not only vitamin C but also antioxidant properties – Line: 142 - 143

Point 4. Please use in the entire manuscript the same abbreviations for the same notions: "f.w." or "f.m." (e.g. in table 1 appears "f.w." and in other parts of manuscript it is used "f.m.").

Response 4. Converted units from f.m. on f.w. – Lines: 156 – 162, 199 - 202

Point 5. In the table 2, the column with Rt (retention time) appear the same values from the next column (wavelength). Please introduce the correct values

Response 5. Valid values have been entered into the table – page 8 - 9

Reviewer 2 Report

The study entitled 'Evaluation of the Chemical Composition of Selected Varieties of L. caerulea var. kamtschatica and L. caerulea var. emphyllocalyx' could be of interest to farmers and researchers in the field, but I feel that there are many weaknesses in the introduction and Materials and Methods. The introduction should include more information on the two botanical varieties and cultivars evaluated and the fruit processing methodology or type.

As far as the method is concerned, the trial should cover at least two years and in any case provide more information regarding the climatic and soil conditions of the two 'nurseries'. Furthermore, the introduction mentions the variability of the antioxidant content depending on the harvest time, why did you not harvest fruits at different ripening periods to identify the best time to harvest the fruit?? Furthermore, I think the experimental design used in the two fields needs to be described.

Author Response

Authors are grateful for the contribution of the Reviewer

Point 1. The study entitled 'Evaluation of the Chemical Composition of Selected Varieties of L. caerulea var. kamtschatica and L. caerulea var. emphyllocalyx' could be of interest to farmers and researchers in the field, but I feel that there are many weaknesses in the introduction and Materials and Methods. The introduction should include more information on the two botanical varieties and cultivars evaluated and the fruit processing methodology or type.

Response 1. More information on the species and fruit processing is given in introduction – Lines 36 – 43, 74 - 85

Point 2. As far as the method is concerned, the trial should cover at least two years and in any case provide more information regarding the climatic and soil conditions of the two 'nurseries'.  

Response 2. More information is provided on the weather conditions and soil conditions of both nurseries – Lines 358 - 365

Point 3. Furthermore, the introduction mentions the variability of the antioxidant content depending on the harvest time, why did you not harvest fruits at different ripening periods to identify the best time to harvest the fruit?? Furthermore, I think the experimental design used in the two fields needs to be described

Response 3.  Experimental design has been described – Lines 358 - 365

Reviewer 3 Report

The manuscript entitled “Evaluation of the chemical composition of selected varieties of L. caerulea var. kamtschatica and L. caerulea var. emphyllocalyx” deals with the fruit chemical composition of different L. caerulea cultivars.

The topic is actual in the field of phytochemistry, thus fitting the Molecules scope, but the purpose of the study remains unclear.

Introduction section should additionally be supported by a literature data of bioactive compounds classes and content in blue honeysuckle.

In Results & Discussion section, there are too many data about different species, which were not the object of the study. Thus, a significant part of the text consists of the results of other authors presenting species that are not related to the investigated object. Therefore, that parts should be compressed and original results should be the focus of the study.

In the manuscript, complete units should be presented. For example, in abstract in the sentence: “The average ascorbic acid content of the L. kamtschatica fruit was 53.5 mg.100 g-1“ unit should be completed as follows: mg.100 g-1 FW, and that should be implemented throughout the manuscript.

In the text, especially in M&M section, original references should be used. If they are modified, this should be specified.

The Conclusion section should be rewritten. In the current form, it is an overview of the results, while the explanations of potential implementation and recommendation for superior cultivar are missing.

Specific comments are given in the pdf file attached.

Author Response

Authors are grateful for the contribution of the Reviewer

The manuscript entitled “Evaluation of the chemical composition of selected varieties of L. caerulea var. kamtschatica and L. caerulea var. emphyllocalyx” deals with the fruit chemical composition of different L. caerulea cultivars.

The topic is actual in the field of phytochemistry, thus fitting the Molecules scope, but the purpose of the study remains unclear.

Point 1. Introduction section should additionally be supported by a literature data of bioactive compounds classes and content in blue honeysuckle.

Response 1. The introduction was supplemented with groups of bioactive compound in fruits – Lines 65 - 68

Point 2. In Results & Discussion section, there are too many data about different species, which were not the object of the study. Thus, a significant part of the text consists of the results of other authors presenting species that are not related to the investigated object. Therefore, that parts should be compressed and original results should be the focus of the study.

Response 2. Data about other species has been compressed – Lines 114 – 119, 137 – 140, 155 – 161, 181 – 186, 189 – 194, 198 – 202, 232 – 239, 264 - 267

Point 3. In the manuscript, complete units should be presented. For example, in abstract in the sentence: “The average ascorbic acid content of the L. kamtschatica fruit was 53.5 mg.100 g-1“ unit should be completed as follows: mg.100 g-1 FW, and that should be implemented throughout the manuscript.

Response 3. Missing units have been added – Lines 19, 153 – 162, 199 - 203

Point 4. In the text, especially in M&M section, original references should be used. If they are modified, this should be specified.

Response 4. Appropriate references have been used – Lines 381, 387, 389

Point 5. The Conclusion section should be rewritten. In the current form, it is an overview of the results, while the explanations of potential implementation and recommendation for superior cultivar are missing. 407- 430

Response 5. The summary has been rewritten, taking into account the information on superior cultivar and potential implementation Lines 407- 429

Notes from PDF files were corrected

Missing reference 1# -  Lines 64 - 69

Compressed text 1# - Lines 114 – 117

Compressed text 2# - Lines 137 – 140

Compressed text 3# - Lines 154 -162

Changed lines for rows – Lines 167, 279, 350

Compressed text 4 # - Lines 198 – 202

Compressed text 5 # - Lines 232 – 239

Reviewer 4 Report

The manuscript corresponding to the ID molecules-2252201 entitled “Evaluation of the chemical composition of selected varieties of L. caerulea var. kamtschatica and L. caerulea var. emphyllocalyx”, deals with characterization of different varieties of L. caerulea. This study is relevant as it provides information about the composition of these berries consumed in Poland, Czech Republic, and Russia, among others, generating interest for consumers and food industry. However, the discussion is scarce as there are lot of studies of these berries, and an improvement in this part will increase the manuscript quality. In this sense, I recommend presenting the values obtained and compare it just with other the studies carried out in Lonicera. It should be better to focus on Lonicera, instead of comparing the results with other berries. Moreover, you should relate your results to those obtained by other authors, instead of simply writing down their results, and explain why these values are important. For example, you can give more importance to the high content of anthocyanins found in the berries, as there are several studies that has demonstrated the ability of these phenolic compounds to… (relate it with health promoting benefits), and present a recent study that evidenced it. You can check this article (DOI: 10.1016/j.fshw.2022.10.013) about benefits of Lonicera in fat absorption inhibition.

Other minor comments:

- Add two decimals in the tables although the last is “0”.

- “In vitro” should be in italics, correct it in the text.

- Tables should be placed at the first time that they are mentioned in the text. The same rules should be applied for figures.

- In line 166, it should be better to write “determined the total polyphenol content of blackberry - 247.25”.

- In table 2, the column Rt is wrong.

- In line 181, it should be better to write “with an amount of 456.3”.

- In line 189, if you do not indicate the total anthocyanin content, the percentages don’t give any information.

- In line 210, indicate that Rutin is Quercetin-3-O-rutinoside.

- In line 219, “with an average concentration of 9.76”. Check it in all text.

- In line 244, you should relate the fact that sorbitol has previously been found in previous studies to your study.

- In line 275, it should be better to write “in triplicate”.

- In line 315, it should be better to write “mean value was 378.4 and 326.68 mg.100 g-1 for L. kamtschatica and L. emphyllocalyx cultivars, respectively”.

- In line 318-319, explain why you choose “Lori” and “Willa” against the other varieties.

Author Response

Authors are grateful for the contribution of the Reviewer

The manuscript corresponding to the ID molecules-2252201 entitled “Evaluation of the chemical composition of selected varieties of L. caerulea var. kamtschatica and L. caerulea var. emphyllocalyx”, deals with characterization of different varieties of L. caerulea. This study is relevant as it provides information about the composition of these berries consumed in Poland, Czech Republic, and Russia, among others, generating interest for consumers and food industry.

Point 1. However, the discussion is scarce as there are lot of studies of these berries, and an improvement in this part will increase the manuscript quality. In this sense, I recommend presenting the values obtained and compare it just with other the studies carried out in Lonicera.

Response 1.  The text has been supplemented with comparisons to other studies - Lines 107 – 109,  130 – 131, 149 – 154, 179 - 181, 186 - 188, 228 – 232, 259 – 264, 287 – 290, 333 - 335, 337- 341

Point 2. It should be better to focus on Lonicera, instead of comparing the results with other berries. Moreover, you should relate your results to those obtained by other authors, instead of simply writing down their results, and explain why these values are important. For example, you can give more importance to the high content of anthocyanins found in the berries, as there are several studies that has demonstrated the ability of these phenolic compounds to… (relate it with health promoting benefits), and present a recent study that evidenced it. You can check this article (DOI: 10.1016/j.fshw.2022.10.013) about benefits of Lonicera in fat absorption inhibition.

Response 2. Comparisons to other species have been shortened and the significance of the obtained results has been emphasized – Lines 114 – 117, 137 - 140, 162 – 165, 198 – 202, 264 – 267,

All minor comments has been rewritten or missing information has been provided:

- Add two decimals in the tables although the last is “0”. – table 1 (page 5), table 3 (page 11)

- “In vitro” should be in italics, correct it in the text. - Line 222

- Tables should be placed at the first time that they are mentioned in the text. The same rules should be applied for figures. - Figure 1 (page 4)

- In line 166, it should be better to write “determined the total polyphenol content of blackberry -  247.25”.- Lines 237

- In table 2, the column Rt is wrong. – page 8 - 9

- In line 181, it should be better to write “with an amount of 456.3”. Line 252

- In line 189, if you do not indicate the total anthocyanin content, the percentages don’t give any information. Lines 254 - 255

- In line 210, indicate that Rutin is Quercetin-3-O-rutinoside. - Line 287

- In line 219, “with an average concentration of 9.76”. Check it in all text. - Line 302

- In line 244, you should relate the fact that sorbitol has previously been found in previous studies to your study. – Line 344 - 346

- In line 275, it should be better to write “in triplicate”. - Line 376

- In line 315, it should be better to write “mean value was 378.4 and 326.68 mg.100 g-1 for L. kamtschatica and L. emphyllocalyx cultivars, respectively”. - Line 415

- In line 318-319, explain why you choose “Lori” and “Willa” against the other varieties. - Line 420 - 425

Round 2

Reviewer 2 Report

I believe the manuscript has been sufficiently improved to be published in Molecules.

Author Response

Authors are greatful for the deep contribution of the Reviewer

Reviewer 3 Report

The revised version of the manuscript is improved, but there are still some issues that should be corrected before publication. Regarding units, value ranges should be written in the same manner: first range in numbers and than units. For example, in Line 134: „1.6 g 100 g-1- 3 g 100 g-1“ should be changed to  „1.6 - 3 g 100 g-1“ and the same throughout the text.

Line 66: „428.14 mg GAE 100 g-1 – 622.52 mg GAE g-1 f.w.“ units are different, which one is correct?

Line 612: results expressed in „f.w.“ are not comparable with those expressed in „d.w.“

There is no need for values in Conclusion section, just major remarks should be included.

Author Response

Authors are greatful for the deep contribution on the Revierwer 

Point 1. The revised version of the manuscript is improved, but there are still some issues that should be corrected before publication. Regarding units, value ranges should be written in the same manner: first range in numbers and than units. For example, in Line 134: „1.6 g 100 g-1- 3 g 100 g-1“ should be changed to  „1.6 - 3 g 100 g-1“ and the same throughout the text.

Response 1. Corrected units for values: Lines – 131, 138, 139, 190, 201

Point 2. Line 66: „428.14 mg GAE 100 g-1 – 622.52 mg GAE g-1 f.w.“ units are different, which one is correct?

Response 2. Wrong unit and appropriate unit value corrected: Line - 65

Point 3. Line 612: results expressed in „f.w.“ are not comparable with those expressed in „d.w.“

Response 3. Another author was quoted: Lines – 333 - 334

Point 4. There is no need for values in Conclusion section, just major remarks should be included.

Response 4. Summary shortened: Lines – 416 - 419